# A robust zirconium amino acid metal-organic framework for proton conduction

Sujing Wang [1], Mohammad Wahiduzzaman [2], Louisa Davis[2], Antoine Tissot[1], William Shepard[3], Jérôme Marrot[4], Charlotte Martineau-Corcos [4,5], Djemel Hamdane[6], Guillaume Maurin [2], Sabine Devautour-Vinot [2] & Christian Serre [1]

Proton conductive materials are of significant importance and highly desired for clean energy-related applications. Discovery of practical metal-organic frameworks (MOFs) with high proton conduction remains a challenge due to the use of toxic chemicals, inconvenient ligand preparation and complication of production at scale for the state-of-the-art candidates. Herein, we report a zirconium-MOF, MIP-202(Zr), constructed from natural α-amino acid showing a high and steady proton conductivity of 0.011 S cm$^{-1}$ at 363 K and under 95% relative humidity. This MOF features a cost-effective, green and scalable preparation with a very high space-time yield above 7000 kg m$^{-3}$ day$^{-1}$. It exhibits a good chemical stability under various conditions, including solutions of wide pH range and boiling water. Finally, a comprehensive molecular simulation was carried out to shed light on the proton conduction mechanism. All together these features make MIP-202(Zr) one of the most promising candidates to approach the commercial benchmark Nafion.

[1] Institut des Matériaux Poreux de Paris, FRE 2000 CNRS, Ecole Normale Supérieure, Ecole Supérieure de Physique et de Chimie Industrielles de Paris, PSL Research Université, 75005 Paris, France. [2] Institut Charles Gerhardt Montpellier, UMR 5253 CNRS ENSCM UM, Université Montpellier, Place Eugène Bataillon, 34095 Montpellier Cedex 05, France. [3] Synchrotron SOLEIL-UR1, L'Orme des Merisiers, Saint-Aubin, BP 48, 91192 Gif-Sur-Yvette, France. [4] Institut Lavoisier de Versailles, UMR 8180 CNRS, Université de Versailles Saint-Quentin-en-Yvelines, Université Paris-Saclay, 78035 Versailles, France. [5] CEMHTI, UPR 3079 CNRS, Orléans Cedex 2 45071, France. [6] Laboratoire de Chimie des Processus Biologiques, Collège de France, 75005 Paris, France. These authors contributed equally: Sujing Wang, Mohammad Wahiduzzaman. Correspondence and requests for materials should be addressed to S.D.-V. (email: sabine.devautour-vinot@umontpellier.fr) or to C.S. (email: christian.serre@ens.fr)

Proton exchange membrane fuel cells (PEMFCs) are among the most promising and attractive candidates for developing clean and renewable energy solutions due to their high energy density, low pollutant emissions, and mild operating conditions[1]. The proton conductive performance is a critical factor for the PEMFC materials and the current commercial benchmark for this application is Nafion[2]. However, the high cost and the eventually decreased performance over cycling of this sulfonated fluoropolymer call for the development of more efficient and cheaper proton conductive materials. Among those, metal-organic frameworks (MOFs) have been reported as a promising class of solids for such application[3]. Nevertheless, so far, the reported MOFs with high proton conduction are hardly environment-friendly to fulfill the sustainable development criteria, due to the involvement of either toxic metal ions or time and effort-consuming organic linker synthesis[4–11]. In this context, the design of highly proton conductive biocompatible MOFs combining excellent stability, low toxicity, and scalable preparation is still a challenging target.

One of the key criteria for biocompatible MOFs is that both inorganic metal and organic linker components show the lowest toxicity. High valence metals with rich natural abundances and long-term stability in their complexes that have been proven to be harmless to human body, such as $Al^{3+}$, $Fe^{3+}$, $Ti^{4+}$, or $Zr^{4+}$, are of prime interests[12,13]. On the other hand, L-α-amino acids which are encoded directly by the triplet codons in the genetic code of DNA are one family of the most attractive naturally occurring linkers suitable for MOFs, not only due to their innocuousness and natural abundant resources but also for their sufficient coordination and bonding abilities towards various metals under a wide range of chemical conditions[14–16]. However, very few examples of MOFs built from pure L-α-amino acids exclusively based on divalent transition metals and main group metals or rare earth metals have been reported so far[17]. Among them, three-dimensional (3D) MOFs with accessible porosity are even scarcer, primarily due to the common chelating coordination fashion of L-α-amino acids[17].

Among the 20 standard L-α-amino acids, aspartic acid is the smallest one consisting of two carboxylate groups with a separation distance sought to be appropriate for energetically favorable robust bridging/linkage of inorganic nodes. A few crystal structures of metal-aspartate MOFs have been reported so far but in all cases they exhibit a limited hydrolytic stability[18–23]. Furthermore, following the Pearson hard and soft acid and base theory, aspartic acid should be appropriate to construct MOFs based on high valence hard transition metal cations while leaving the soft amino group free from coordination. However, no example of hard metal cations, especially $Al^{3+}$, $Fe^{3+}$, or $Zr^{4+}$, has been reported to form MOFs based on amino acids up to now, possibly due to the fast and strong bonding between high valence metal ions with aspartate in solution which leads highly complex the generation of crystalline products[24,25].

It is worth mentioning that the porous Zr-amino acids MOF would be of great interest since in general high stability and excellent performance in various applications have been highlighted for Zr-carboxylate MOFs over the last few years[26]. In the field of proton conductive materials, the design of Zr-MOFs based on non-commercially available linkers has been actively investigated, including a flexible tetraphosphonate coordination polymer[27], a Zr-phosphonate based on glyphosine[28,29], a phenolate-based MOF[30], and PCMOF20[31], outperforming in some cases Nafion. However, these MOFs suffer from either a lack of biocompatible character, the use of expensive or poorly environment-friendly chemicals and/or scale-up limitations. Series of Zr-carboxylate MOFs based on commercial (not bio-derived) linkers such as the Zr-terephthalate UiO-66 as well as the Zr-trimesate MOF-808[32] have also been studied as proton conductive materials, in particular the UiO-66's series of functionalized solids that exhibit interesting performances being tuned by (i) the decoration of the pore wall with pendent acid groups ($-CO_2H$, $-SO_3H$)[4,33–36] or (ii) the incorporation of ligand defects saturated or not by acid treatment[37–39]. Some of these solids are however not exceeding the performances of Nafion.

We report herein an amino acid-based Zr-MOF, denoted as MIP-202(Zr) (MIP stands for the Materials of the Institute of porous materials from Paris), in which the 12-connected $Zr_6(\mu_3-O)_4(\mu_3-OH)_4$ node and the L-aspartate spacer are assembled in water generating a 3D microporous framework with a UiO-66[40] type structure. MIP-202(Zr) demonstrates excellent proton conductivity up to 0.011 S cm$^{-1}$ at 363 K and under 95% of relative humidity (RH), with the crystal structure unchanged even after one week of test. This makes MIP-202(Zr) one of the best proton conductive MOF materials reported so far. The proton conduction mechanism was further investigated by means of molecular simulations. In addition, MIP-202(Zr) not only features green and scalable synthesis, but also is the only amino acid-based MOF material that possesses a very good hydrolytic stability and chemical stability.

## Results

**Synthesis and characterization.** The syntheses of most Zr-MOFs involves toxic organic solvents, such as dimethylformamide and dimethylacetamide, carried out in sealed reactors. Additionally, diluted solutions or suspensions of starting chemicals are generally preferred in order to yield Zr-MOF product with good crystallinity[26]. Thereby, it is hardly a scalable and environment-friendly preparation process, which is undoubtedly against the global sustainable development criteria[41,42]. Moreover, all the best proton conductive MOFs ($\sigma > 10^{-2}$ S cm$^{-1}$) have been obtained under harsh conditions so far[11]. The scale-up of those MOFs would therefore face severe problems such as the use of toxic and expensive chemicals, complicated linker synthesis, and reproduction issues (Supplementary Table 1). In sharp contrast, MIP-202(Zr) was synthesized by simply heating the mixture of $ZrCl_4$ and L-aspartic acid in water for several hours under reflux with ambient pressure. Besides, the activation step only involves green solvents, such as water and ethanol, for washing and removing the excess of free linker and Cl$^-$ extra-framework species. This approach is not only environment-friendly, but also highly beneficial in terms of cost and regeneration. This procedure also results in a high space-time yield of 7030 kg m$^{-3}$ day$^{-1}$ comparable to some industrial scale syntheses[43,44].

It is worthy to note that only high concentration of reactants could generate the MIP-202(Zr) product efficiently. On the contrary, diluted solution of the same starting materials resulted in clear solution without any solid product under the same reaction condition. It could be due to the strong acidic condition associated with $Zr^{4+}$ in solution that affects the presenting conformation of the amino group containing linkers. It has been documented before for UiO-66(Zr)-NH$_2$[45] that a partial protonation of the weakly basic amino group occurs as a consequence of the highly acidic synthesis conditions that significantly slows down the kinetic of formation of this MOF in comparison with other functionalized UiO-66. Indeed, if one considers amino acids that possess much more basic $-NH_2$ groups, this is likely that a full protonation of the amino group might occurs in the presence of strongly acidic metal cations, leading to zwitterionic ligand species, which certainly alters the ability of the amino acid to complex the high valence metal species. Thus, in the case of MIP-202(Zr), the full protonation of the amino group on the L-aspartic acid molecule is likely to slow down the crystallization process

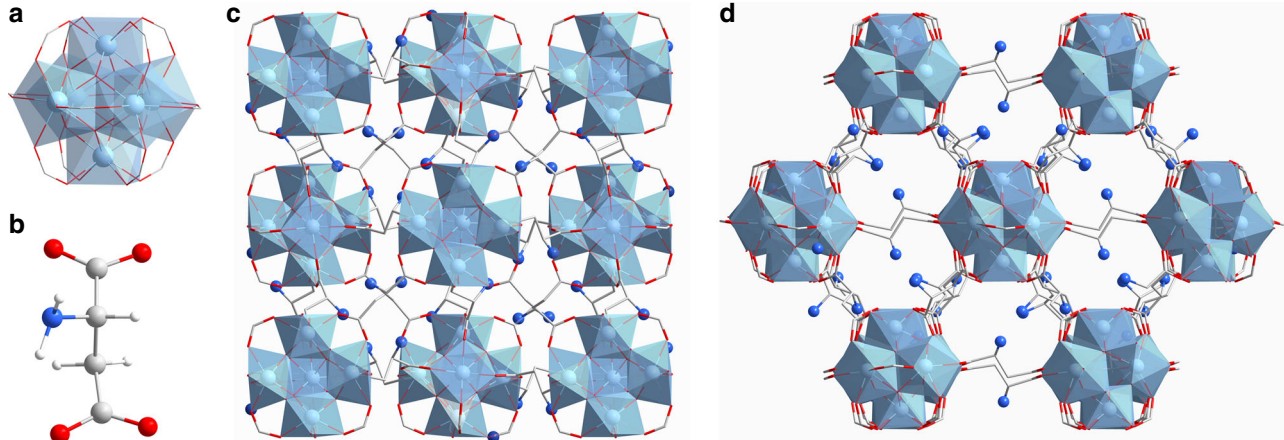

**Fig. 1** Crystal structural features of MIP-202(Zr). **a** The 12-connected $Zr_6(\mu_3\text{-}O)_4(\mu_3\text{-}OH)_4(COO^-)_{12}$ cluster SBU. **b** An aspartic acid linker. **c** The crystal structure of MIP-202(Zr) viewed along the *a*-axis. **d** The crystal structure of MIP-202(Zr) viewed along the (101) plane. Zr atoms or polyhedra, oxygen, carbon, nitrogen, and hydrogen atoms are in light blue, red, gray, dark blue, and white, respectively

through the formation of stable highly polarized metal complexes in solution. Only when considering over saturated conditions, one can successfully shift the equilibrium towards the formation of precipitates and thus MIP-202(Zr).

Furthermore, the MIP-202(Zr) product particle size distribution can be tuned easily. For example, reaction without stirring leads to single crystals with a size range of 5–30 μm (Supplementary Figure 1), which is suitable for single crystal X-ray diffraction data collection under synchrotron radiation, easy recovery of the material during the filtration process to collect the MOF after the synthesis and/or activation/cleaning in solution as well. On the contrary, reactions with stirring under reflux result in much smaller particle size of the MOF product down to sub-micro level which can be advantageous to prepare thin films or membranes (Supplementary Figure 2). Consequently, this opens the way towards the use of MIP-202(Zr) in practical applications.

The crystal structure of MIP-202(Zr) was solved by employing our newly developed crystal structure prediction software which is based on the Automated Assembly of Secondary Building Units (AASBU)[46] method (Supplementary Note 1 and Supplementary Figure 3). The structural framework was experimentally confirmed by synchrotron diffraction data collected with a micro-focused X-rays on the Proxima 2A beamline (Synchrotron SOLEIL, France)[47] using a microsized single crystal (Supplementary Note 2). It should be noted that MIP-202(Zr) is isostructural to the previously reported cubic (fcu topology) Zr-Fumarate (also known as MOF-801) compound (Fig. 1)[48,49]. Hence their Powder X-ray Diffraction (PXRD) patterns match with each other very well (Supplementary Figure 4), resulting in almost identical unit cell parameters—a = 17.8348(17) Å and 17.8260(21) Å for Zr-Fumarate and MIP-202(Zr), respectively—consistent with the similar molecular lengths of fumaric acid and aspartic acid. The uncoordinated amino groups are highly disordered over four positions. It is interesting to note that all the amino groups are pointing towards the pores, making the cavity environment of MIP-202(Zr) very different from that of the Zr-Fumarate compound.

Scanning electron microscopy coupled with energy-dispersive X-ray spectroscopy (SEM-EDX) evidenced that there are still $Cl^-$ species trapped inside the porosity of MIP-202(Zr). Extensive washing the MOF sample in boiling ethanol or methanol over prolonged periods of time could effectively remove most of the free $Cl^-$ residue, leading to a ratio of $Cl^-$ to both Zr and N of 1:1. This suggests that all the amino groups of the aspartate linker molecules are protonated and are present in the form of $-NH_3^+/$

$Cl^-$ pair (Supplementary Table 2). The solid-state $^{15}N$ magic-angle spinning (MAS) NMR spectrum (Supplementary Figure 5) collected on the MIP-202(Zr) sample supports the presence of the above-mentioned $-NH_3^+/Cl^-$ pair in the MOF structure since the chemical shift of the nitrogen species in MIP-202(Zr) is very close to that of the amino group in a zwitterionic form[50]. Therefore, the $NH_3^+$ moieties in the aspartate ligands are considered along with $Cl^-$ counter ions for the final structure optimization at the density-functional level of theory (DFT). It is important to note that the presence of this $-NH_3^+/Cl^-$ pair plays a critical role on the accessibility of the MIP-202(Zr) porosity. As expected from the crystal structure, nitrogen sorption result collected on the fully protonated MIP-202(Zr) sample leads to a limited accessible pore volume of 0.1 $cm^3\,g^{-1}$ and a free pore diameter less than 4 Å which falls into the range of ultra-microporous materials class (Supplementary Figures 6–8).

Since L-aspartic acid was applied as the linker to synthesize MIP-202(Zr), circular dichroism (CD) measurements were carried out to access its possible chirality. MIP-202(Zr) constructed from D-aspartic acid was prepared as well under the same reaction condition for comparison. After decomposition of the MOFs structures in concentrated NaOH solution, CD spectra were collected on the clear solutions containing released aspartate from the MOF frameworks. As expected, the digested MIP-202(Zr) samples display nearly the same CD response as that obtained from the enantiomers of aspartic acid and pre-prepared Na-aspartate salts under the same testing condition (Supplementary Figure 9), indicating the chiral pore environments of both L-MIP-202(Zr) and D-MIP-202(Zr) samples.

Of an utmost importance for practical applications, particularly in the case of proton conductivity, the hydrothermal stability of MOFs is prerequisite. Various conditions were applied to the MIP-202(Zr) sample in order to confirm its expected chemical resistance. MIP-202(Zr) not only shows an excellent stability in common organic solvents (e.g. methanol) as the other reported aspartate-based MOFs[17] but also displays remarkably elevated hydrolytic stability in water. Boiling water did not affect at all the crystallinity of MIP-202(Zr) sample as supported by PXRD and nitrogen sorption measurement results (Supplementary Figures 7, 8 and 10a). Furthermore, the long-range order of the MIP-202(Zr) crystalline framework is stable over a large pH range in aqueous solutions from acidic (pH = 0, HCl) to basic (pH = 12, KOH) conditions as confirmed by PXRD. The increased nitrogen uptakes and pore sizes for the samples exposed to acidic and basic conditions are due to the removal of $Cl^-$ counter ions or the

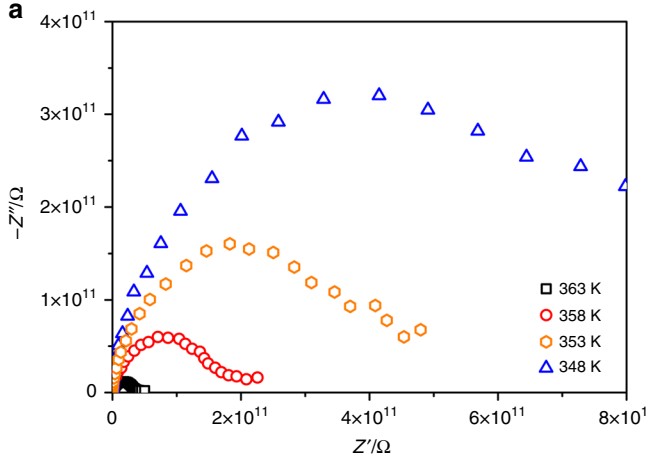
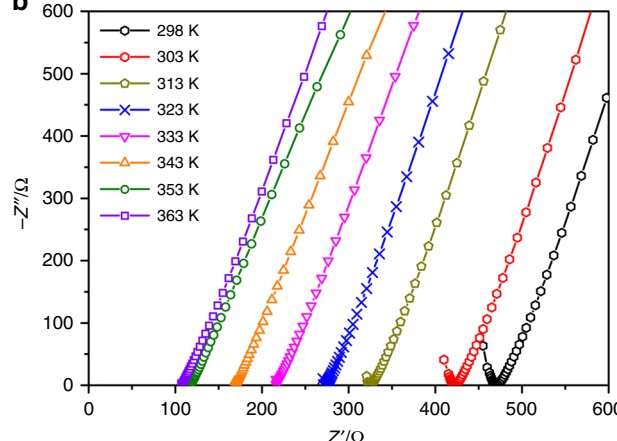

**Fig. 2** Nyquist plots from AC impedance data for MIP-202(Zr). **a** at 0% RH. **b** at 95% RH

generation of structural defects respectively, as evidenced by a combined analysis of TGA, EDX and elemental analysis results (Supplementary Figure 11, Supplementary Tables 3 and 4, Supplementary Note 3), with more defects being created under acidic treatment or a larger amount of Cl anions removed under basic conditions. The overall chemical stability of MIP-202(Zr) is therefore comparable to or to some extent even better than most of the previously reported Zr-MOFs based on multi-topic aromatic carboxylic linkers. In addition, to the best of our knowledge, MIP-202(Zr) is the first crystalline metal-amino acid-based porous solid material, MOFs or otherwise, that not only displays an excellent stability in the presence of water but also a remarkable resistance to both acidic and basic conditions.

**Proton conductivity**. MIP-202(Zr) can be considered as a potentially efficient proton conductor based on its following features prerequisite for such application: (i) the presence of accessible $NH_3^+$ Brønsted acidic groups (proton sources) that decorate the short aliphatic chain of the aspartate linker, (ii) the crystalline tunnel-like architecture and the small porosity that is expected to favor the formation of an oriented hydrogen-bonded network of the guest water molecules, and (iii) a good hydrolytic stability to ensure that the proton conductive performances can be maintained over cycles. AC impedance spectroscopy was employed to assess the performance of MIP-202(Zr) (Supplementary Note 4). Measurements were firstly carried out on the anhydrous solid, which was pre-treated under nitrogen flow at 363 K for 2 h. Figure 2a illustrates the Nyquist plot as a function of temperature for the anhydrous MIP-202(Zr) (RH = 0%). The semi-circle represents the bulk and grain boundary resistances, and the small tail at low frequency is indicative of the accumulation of ionic charges at the blocking electrode interface. It is worthy to note that the tail is poorly pronounced in relation with the very low conductivity values recorded at temperature lower than 363 K ($\sigma \leq 10^{-11}$ S cm$^{-1}$; Supplementary Figure 12 and Supplementary Table 5). This observation evidences that the charge carriers are poorly mobile in the absence of water. In contrast, the evolution of the imaginary ($Z''$) part of the impedance vs the corresponding real part ($Z'$) drastically changes for the material equilibrated for 24 h under 95% RH (Fig. 2b). It results in a sharp increase of the conductivity by nine orders of magnitudes compared to the anhydrous state, leading to the value of $1.1 \times 10^{-2}$ S cm$^{-1}$ under the condition of 363 K/95% RH (Supplementary Figure 13). This value is among the highest conductivities reported for other water-mediated proton-conducting MOFs, including UiO‐66-

$(SO_3H)_2$[34], $(Hpy)_2[Zn_2(ox)_3].nH_2O$[51], VNU-15[52], JLU-Liu44[53], Fe-CAT-5[5], PCMOF2$^1/_2$[8], and PCMOF20[31], and converges towards the performances of the benchmark Nafion based polymer[54]. Furthermore, MIP-202(Zr) is the best proton conductor among the MOFs showing a cationic framework, such as the Zn-(4-carboxyphenyl)imidazolium-based solid with a $\sigma = 2.3 \times 10^{-3}$ S cm$^{-1}$ at 298 K/95% RH[55], and the Cu-bipyridine glycoluril (BPG) based MOF with a $\sigma = 4.4 \times 10^{-3}$ S cm$^{-1}$ at 363 K/95% RH[56]. In addition, the conductive performances of MIP-202(Zr) are well maintained over one week of experiments performed at 363 K and 95% RH (Supplementary Figure 14). This observation proves the robustness of MIP-202(Zr) under the operating conditions, as supported by the analysis of the PXRD pattern of the solid after the conductivity measurements, which confirms the integrity of the crystal structure during the impedance analysis (Supplementary Figure 10b). Finally, the variation of the conductivity performance with temperature was also studied at RH = 95% from 363 K to 298 K (Supplementary Table 6). The activation energy, deduced from the linear fitting of ln($\sigma$T) vs 1000 T$^{-1}$ according to the Nernst–Einstein equation $\sigma(T) = \sigma_0/T$ exp $(-Ea/kT)$, is 0.22 eV (Supplementary Figure 15). This value suggests that the proton diffusion is governed by a Grotthus mechanism involving proton hops between hydrogen-bonded water molecules (0.1–0.4 eV).

**Molecular simulation**. Monte Carlo (MC) calculations (Supplementary Note 1, Supplementary Figure 16 and Supplementary Tables 7–9) were further performed on the fully hydrated MIP-202(Zr) to analyze the preferential arrangements of the confined water molecules and the roles of both Cl$^-$ and $NH_3^+$ in the formation of the hydrogen-bonded network (Supplementary Figures 17–20). Such feature is a key to allow the proton propagation at the origin of the promising proton conductive performance of MIP-200(Zr). The analysis of the MC configurations indicated that $H_2O$ molecules are arranged in such a way that they interact with the MOF framework through their oxygen atoms ($O_w$) with (i) the proton of the $NH_3^+$ group of the linker as evidenced by a short mean distance between the two atoms pairs [d($O_w$–$H_{NH3}$) = 1.8 Å] (Fig. 3a) and (ii) the proton of the $\mu_3$-OH group of Zr oxocluster with a mean distance d($O_w$–$H_{\mu3\text{-}OH}$) of 1.9 Å (Fig. 3b), resulting in a well-ordered hydrogen-bonded network. Moreover, the confined water molecules themselves also form a 3D hydrogen-bonded network corresponding to the $H_w$–$O_w$ mean distance of 1.8 Å (Fig. 3d), as short as those usually observed for water in the bulk state[57]. This spatial distribution and the interactions at play are reminiscent of those commonly

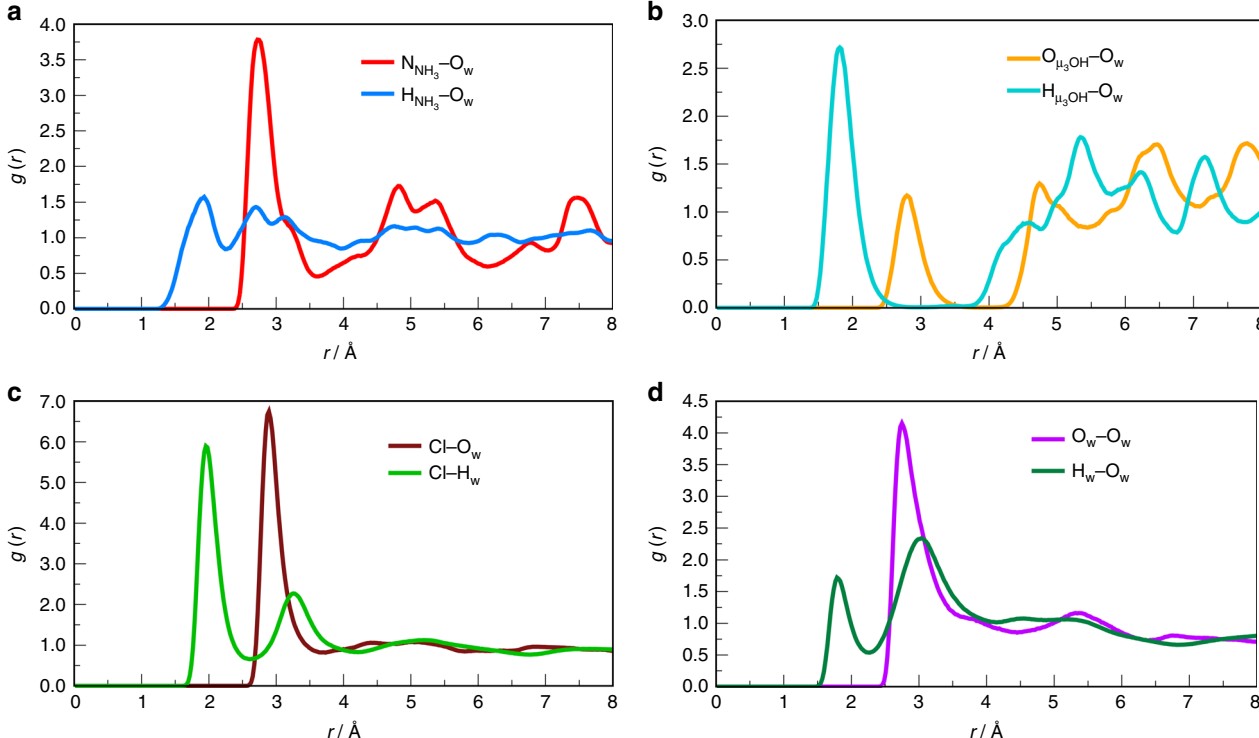

**Fig. 3** Radial distribution functions calculated from the average over the MC configurations considered for hydrated MIP-202(Zr) at 90 °C. **a** N and H of $NH_3^+$ to $O(H_2O)$ atomic pairs. **b** O and H of $\mu_3$-OH to $O(H_2O)$ atomic pairs. **c** Cl to O and H of $H_2O$ atomic pairs. **d** O and H of $H_2O$ to $O(H_2O)$ atomic pairs

reported for water-mediated proton-conducting MOFs[58]. Such predictions are consistent with the existence of a Grotthus-like mechanism, supported by the value of the activation energy estimated from the conductivity measurements (0.22 eV) which is far below 0.4 eV.

A quantitative analysis of the hydrogen bond number per $H_2O$ molecule was calculated using the following criteria: the distance between donor and acceptor oxygen centers (D−A) is shorter than 3.5 Å and the corresponding angle between the intramolecular O−H vector (OH of donor molecule) and the intermolecular O−O vector (O atoms of donor to acceptor) is less than 37°. These criteria are the same as we used previously to describe the hydrogen bonds in other porous materials[35,36,58]. The average number of hydrogen bond formed by a single $H_2O$ molecule over all the generated MC configurations is 4. Implying the same hydrogen bond definition for the $NH_3^+$ group (N as a donor, O of water as an acceptor and H of $NH_3^+$ as a proton), a single N donor exhibits maximum three hydrogen bonds with neighboring water molecules. Such hydrogen bonding extends in a way that forms water bridge with neighboring $NH_3^+$ moieties, simultaneously generating a −NH⋯:O:⋯HN− (where O is the oxygen of a bridging water) network (Fig. 4b). This strongly suggests that a hydrogen-bonded network can be initiated from the $NH_3^+$ functional groups and creates possible pathways for the proton movement throughout the framework. In addition, Cl⁻ ions present within the cavities also interact with protons and a large fraction of them fulfill the aforementioned hydrogen bonding criteria by taking the acceptor positions around $NH_3^+$ and $H_2O$ molecules (Fig. 4c). Some of these anions form single hydrogen bond with $NH_3^+$ or $H_2O$, disrupting the potential hydrogen-bonded network in certain directions. In some cases, a single Cl⁻ holds a position that favors multiple hydrogen bonds was noticed. Thus we assume that such an arrangement is not necessarily detrimental for the proton propagation.

We further performed quantitative analysis for the clustering of water molecules within the pores of MIP-202(Zr). A cluster is defined as a continuous network of water molecules interconnected by hydrogen bonds among themselves. We revealed that the adsorbed water molecules form hydrogen-bonded aggregates incorporating up to 10 $H_2O$ molecules, this cluster-size being slightly smaller than those observed for other MOFs with 1D channels, such as MIL-53(Cr) and MIL-47(V)[59]. Such arrangements of water make bridges among multiple proton sources, i.e., $NH_3^+$ groups present in the adjacent or opposite pore walls, and this leads to a much more extended hydrogen-bonded network connection (Supplementary Figure 21). Indeed, the existence of such pathways offers an optimal situation for the water-mediated transport of protons over long distances, bridging different aspartate ligands that act as proton sources in MIP-202(Zr).

In summary, MIP-202(Zr), a Zr-MOF based on natural amino acid linker has been successfully synthesized and fully characterized. Unlike most of the previously reported proton conductive MOFs, MIP-202(Zr) features a green, scalable and facile preparation by using cost-effective and environment-friendly chemicals. This material displays an excellent hydrolytic and chemical stability in aqueous solutions with a wide range of pH. Our results therefore pave the way for the design of green, robust, and cheap MOFs based on amino acids. It is worth mentioning that MIP-202(Zr) features outstanding proton conduction performances under humid condition that are driven by the formation of an extended hydrogen-bonded network resulting from the interactions between the $NH_3^+$ groups acting as the proton source, and the water molecules present in the pores. To the best of our knowledge, MIP-202(Zr) is one of the most practical alternative materials to the commercial benchmark Nafion for ion-exchange membrane applications. Furthermore, the tunable product particle size, the pore system with chiral environment in the structure, and the bio-compatibility make

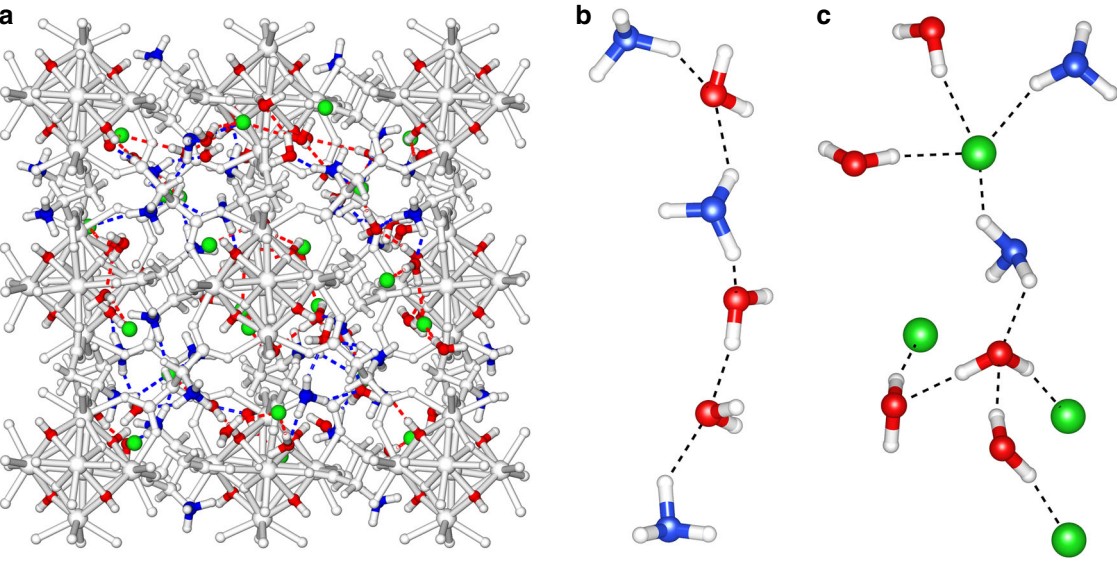

**Fig. 4** Illustration of the hydrogen bonding in MIP-202(Zr) structure. **a** Multiple water-mediated hydrogen-bonded network as evidenced by MC-NVT simulations performed at 90 °C for hydrated MIP-202(Zr). **b** A representative hydrogen-bonded water-bridge network formed by the NH$_3$ and H$_2$O molecules. **c** Hydrogen bonding with the possible involvement of Cl$^-$ ions (MOF skeleton in gray, nitrogen of NH$_3^+$ group in blue, Cl$^-$ ions in green and oxygen of $\mu_3$-OH, and H$_2$O molecule in red)

MIP-202(Zr) as a potentially promising candidate in biological and pharmaceutical applications.

## Methods

**Typical synthesis of MIP-202(Zr)**. L-Aspartic acid (2.8 g) was transferred into a 25 mL round bottom flask; 5 mL of water was added to disperse the linker. ZrCl$_4$ (2.33 g) was added by portion into the water suspension, leading to a clear colorless solution. Another 5 mL of water was added to flush the inner face of the flask, trying to wash as much as possible the ZrCl$_4$ powder down into the solution. The reaction was kept at reflux for 1 h (120 °C). After cooling down to room temperature, the expected product of MIP-202(Zr) was collected by filtration, washed with water/EtOH at room temperature, and air dry, leading to 2.93 g white solid product of the MOF with a crude yield of 90%. This preparation method could be applied to either smaller scale syntheses or larger scale ones when a reflux apparatus with suitable size is involved. For example, a scale-up synthesis was carried out in a 250 mL round bottom flask under the same condition results in 30.4 g of product (94% yield) with a space-time yield of 7296 kg m$^{-3}$ day$^{-1}$.

**Characterization**. Powder X-ray diffraction (PXRD) data were recorded on a high-throughput Bruker D8 Advance diffractometer working on transmission mode and equipped with a focusing Göbel mirror producing CuKα radiation ($\lambda = 1.5418$ Å) and a LynxEye detector. Nitrogen porosimetry data were collected on a Micromeritics Tristar instrument at 77 K. SEM-EDX results were recorded with an FEI Magellan 400 scanning electron microscope. TGA data were collected on Mettler Toledo TGA/DSC 2, STAR System apparatus with a heating rate of 2 °C/min under the oxygen flow. Elemental analyses were performed on a Vario EL-III elemental analyzer. Solid-state NMR spectra were recorded with an Advance Bruker 500 NMR spectrometer. Circular dichroism (CD) spectra were recorded on a Chirascan™-plus CD Spectrometer (Applied Photophysics). The far ultraviolet spectra (185–260 nm) were measured at 20 °C in quartz cells of 0.4 cm optical path length. Spectra were acquired at a resolution of 1 nm, with time per points set at 0.7 s and a bandwidth of 1 nm.

**Proton conductivity measurement**. Impedance measurements were performed on a Broadband Dielectric Spectrometer, Novocontrol alpha analyzer over a frequency range from 1 Hz to 1 MHz with an applied ac voltage of 20 mV. Measurements were collected on the anhydrous and the hydrated solid, which was placed into an Espec Corp. SH-221 incubator, to control the temperature (298 < $T$ (K) < 363) and the RH (95%). The solid was equilibrated for 24 h at given $T$ and RH values to ensure fixed water content before recording the impedance. The measurements were performed using samples as a powder, introduced between two gold electrodes in a parallel plate capacitor configuration with an annular Teflon spacer for insulation, allowing the use of the two-probe method for the electrical measurements. The thickness $l$ and surface $S$ are as follows: $l = 1.11$ and 2.07 mm, $S = 33.1$ and 15.9 mm$^2$ for the anhydrous and the hydrated MIP-202(Zr), respectively. Conductivity was deduced from the Bode diagram as well as from the Nyquist plot for comparison.

**X-ray crystallography**. The single crystal data of MIP-202(Zr) were collected on Proxima-2A microfocused beamline at the SOLEIL Synchrotron ($\lambda = 0.70846$ Å) at 100 K. The data were integrated and reduced using XDS software. The structure was then solved using SHELXT and refined with SHELXL software. The compound MIP-202(Zr) is isostructural to the previously reported cubic (fcu topology) Zr-Fumarate (MOF-801) compound. The aspartic acid linkers are highly disordered in the structure. Therefore, the nitrogen atom from the linker is located on four distinct positions with partial occupations (noted N1 and N2), corresponding to the four possible orientations of the linker. The central carbon atoms (noted C2) from the linker should also be disordered on two distinct positions. However, due to symmetry considerations and to the limited quality of the single crystal diffraction data, only the average position of these atoms has been determined.

## Data availability

The X-ray crystallographic data for MIP-200(Zr) have been deposited at the Cambridge Crystallographic Data Centre (CCDC), under deposition number CCDC 1842337. These data can be obtained free of charge from the CCDC via www.ccdc.cam.ac.uk. All other relevant data supporting the findings of this study are available from the corresponding authors on request.

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

## Acknowledgements

S.W., G.M., and C.S. acknowledge the financial support of the European Community within the Seventh Framework Programme (FP7) under Grant Agreement No. 608490 (Project M4CO2). C.M. and G.M. are grateful for financial support from Institut Universitaire de France (IUF).

## Author contributions

S.W. contributed to the synthesis and general characterization of MIP-202(Zr). M.W. and G.M. performed the computational assisted structure determination of MIL-202(Zr) and the modeling of the proton migration mechanism. L.D. and S.D.-V. carried out the proton conductivity data collection. A.T., J.M., and W.S. collected the synchrotron diffraction data on the single crystal and solved the crystal structure. C.M.-C. conducted the solid-state NMR characterizations. D.H. collected the CD data. C.S. closely supervised the

synthesis and characterization part of the work. C.S. and S.D.-V. co-coordinated the entire study. All authors contributed to the writing and discussion.

## Additional information

**Competing interests:** The authors declare no competing interests.

