## [Peer Review File · Nature Communications]

Reviewers' comments:

Reviewer #1 (Remarks to the Author):

The manuscript by Wang et al. reports on the water-based synthesis and proton conduction properties of a new zirconium-based metal-organic framework (MOF) containing the naturally occurring aspartic acid as the organic linker, named MIP-202(Zr). The compound displays a cubic framework, practically isostructural with that of MOF-801 (based on fumaric acid as a linker), with protonated amino groups exposed in the pores. The presence of these functional groups makes MIP-202(Zr) an excellent proton conductor, ranking among the very best MOFs reported to date. In addition, the compound shows excellent stability in working conditions, with no loss of conductivity over a period of one week and no apparent loss of crystallinity, which is an element of great interest for practical application. Monte-Carlo simulations provide insight into the hydrogen bond network involving -NH_3^+ groups, Cl^- , -OH groups from the clusters and H_2O molecules within the porous structure and the proton transport mechanism. The manuscript is well written and technically sound, and these results are well worth of being published in Nature Communications.

I do have concerns about some aspects of the work though, which the authors should address prior to publication:

- MIP-202 was treated in various conditions to demonstrate its stability. This conclusion is mainly drawn from PXRD, that shows that no loss of crystallinity occurs. However, N_2 sorption analysis displays that the uptake significantly increases (up to 2 times the original uptake) after the treatment, especially in acidic and alkaline conditions. In my opinion, this should not be overlooked. The pK_a of the amino group of aspartic acid is about 9.5, meaning that at pH 12 (and possibly even at pH 10) this should be deprotonated, with consequent removal of Cl^- from the pores. This would free up a significant amount of space within the pores, as the result of the analysis seems to suggest, and have an effect on the physical-chemical character of the material, including the proton conduction properties of the material. This is something that I deem worth of being investigated in order to better understand the system. What is less straightforward to interpret is the large increase in N_2 adsorption after treatment at pH 1-3. My guess is that defects could be formed upon exposure to acidic conditions and the authors should make an effort in trying to rationalise this behaviour as well.
- In connection with the above, pore size distribution analysis should be carried out to better characterise the framework. The only information on this aspect is derived from calculations, but experimental evidence is also needed. The N_2 isotherms available should be enough to determine pore size distribution down to about 10 Angstrom diameter, which could already be useful to see differences before and after treatment.
- Evident differences also in the TGA curves arise after treatment, but the way the curves are currently drawn does not help to appreciate them. I recommend to plot the TG curves normalising to the formula weight of ZrO_2 as 100%, which is the likely decomposition product formed at 600 degrees, and to discuss these results more carefully.
- The authors determine the Cl/Zr ratio using EDX, whilst they assume that the Cl/N ratio is 1. I believe that this is not sufficient to have a correct chemical knowledge of the compound and elemental analysis should be performed to determine the CHN content.

Minor points are the following:

- In the introduction (Page 3, Lines 51-54) the authors state "Nevertheless, so far, the reported MOFs with high proton conduction are hardly environment-friendly to fulfill the sustainable development criteria, due to the involvement of either toxic metal ions or time and effort consuming organic linker synthesis.". References should be added to support this statement.
- In the caption for Figure 1, the SBU formula is wrongly reported as " $\text{Zr}_6(\mu_3\text{-O})_4(\mu_3\text{-OH})_4(\text{COOH})_{12}$ ". The carboxylic groups should be deprotonated in order to ensure electroneutrality. Please correct.
- At Page 5, Line 105, it stated that the synthesis of most Zr-MOFs is carried out in solvothermal

conditions. I do not agree with this terminology, because Zr-MOFs are usually synthesized at temperatures below the normal boiling point of the amide solvent. Since no autogeneous pressure is developed, this does not classify them as properly solvothermal.

- At Page 7, line 136, the expression "under synchrotron beamline" should be changed to "under synchrotron radiation".

- At Page 7, Line 152, it is stated "The uncoordinated amino groups are highly distorted over four positions." The term "distorted" should be changed into "disordered".

Reviewer #2 (Remarks to the Author):

The authors report a green and efficient method to synthesize the MIP-202 (Zr) constructed from natural α -amino acid. The Zr-MOF shows outstanding proton conductive performance. Besides, the authors also use quantitative analysis to explore the mechanisms. These features bestow this novel material with good application potential. Therefore, I would like to recommend the publication of this manuscript with the following revisions. Major revision is suggested.

1. Line 116, the authors claim "This procedure also results in a high space-time yield of 7030 kg m⁻³ day⁻¹ comparable to some industrial scale syntheses". However, no scalable preparation results could be found in this paper. More evidence should be provided since the large-scale production process might have a big difference with the lab-scale preparation.

2. Line 203, the authors point out that the formation of orientated H-bond network in MOFs. Please provide some experimental evidence.

3. Line 204, the authors claim "a good hydrolytic stability to ensure that the proton conductive performances can be maintained over cycles." Could the material keep stability under redox conditions? Maybe Fenton's test should be conducted.

4. Quantitative analysis are used to explore the mechanism of high proton conductivity of MIP-202 (Zr). The simulation results would become more convincing if they are verified or partially verified by experimental results.

5. Line 282, the authors claim "The adsorbed water molecules are only able to form small discontinuous aggregates of hydrogen-bonded clusters within the pores of MIP-202(Zr)". Will these discontinuous aggregates hydrogen-bonded clusters confine the long-range proton conduction?

Reviewer #3 (Remarks to the Author):

The work reports the synthesis, structural stability study, proton conductivity, and MC results of a MOF. The MOF was stable under several solvent conditions and acidic or basic media. This material was also prepared under mild reaction conditions. Proton conduction of the MOF was investigated experimentally and theoretically. However, the conductivity was just as high as 0.01 S cm⁻¹, which is not significant compared to the state-of-the art conductivity (>0.1 S cm⁻¹) observed in several MOFs. Therefore, I think this paper is not suitable for this journal. The following concerns should be addressed.

1. The dinitrogen sorption data (Fig. S3) showed that all treated samples have adsorption capacities greater than as-made sample. This indicates that the chemical treatments affect the porosity in spite of structural integrity. The authors should discuss what happens.

2. To support the Grotthuss mechanism, additional experiments were added to manuscript.

3. Calculate the yield of MIP-202(Zr)

4. Check typos in the manuscript. For instance, Table 1 in line 112 should be changed to Table 2.

Reviewers' comments:

Reviewer #1 (Remarks to the Author):

“The manuscript by Wang et al. reports on the water-based synthesis and proton conduction properties of a new zirconium-based metal-organic framework (MOF) containing the naturally occurring aspartic acid as the organic linker, named MIP-202(Zr). The compound displays a cubic framework, practically isostructural with that of MOF-801 (based on fumaric acid as a linker), with protonated amino groups exposed in the pores. The presence of these functional groups makes MIP-202(Zr) an excellent proton conductor, ranking among the very best MOFs reported to date. In addition, the compound shows excellent stability in working conditions, with no loss of conductivity over a period of one week and no apparent loss of crystallinity, which is an element of great interest for practical application. Monte-Carlo simulations provide insight into the hydrogen bond network involving -NH^{3+} groups, Cl^- , -OH groups from the clusters and H_2O molecules within the porous structure and the proton transport mechanism. The manuscript is well written and technically sound, and these results are well worth of being published in Nature Communications.”

We thank the reviewer for his/her positive evaluation of the paper.

“I do have concerns about some aspects of the work though, which the authors should address prior to publication:

1. MIP-202 was treated in various conditions to demonstrate its stability. This conclusion is mainly drawn from PXRD, that shows that no loss of crystallinity occurs. However, N_2 sorption analysis displays that the uptake significantly increases (up to 2 times the original uptake) after the treatment, especially in acidic and alkaline conditions. In my opinion, this should not be overlooked. The pKa of the amino group of aspartic acid is about 9.5, meaning that at pH 12 (and possibly even at pH 10) this should be deprotonated, with consequent removal of Cl^- from the pores. This would free up a significant amount of space within the pores, as the result of the analysis seems to suggest, and have an effect on the physical-chemical character of the material, including the proton conduction properties of the material. This is something that I deem worth of being investigated in order to better understand the system. What is less straightforward to interpret is the large increase in N_2 adsorption after treatment at pH 1-3. My guess is that defects could be formed upon exposure to acidic conditions and the authors should make an effort in trying to rationalise this behaviour as well.”

Response:

We thank this referee for these constructive comments. We collected extra characterization data on the samples exposed to acidic and basic conditions including TGA, EDX and elemental analysis. A careful analysis of this whole set of data confirmed the statement made by the reviewer. Indeed, the treatments in alkaline conditions led to the removal of a considerable amount of trapped Cl^- in the pore while only a small concentration of missing linkers in the MOF structure, i.e. local structural defects, (2-3% molar ratio) was created. The removal of Cl^- is thus the origin of the increase of the N_2 uptake. When the sample was treated with HCl , we observed

the creation of a larger number of missing linker defect in the MOF structure (7% and 10% for HCl concentrations of 1×10^{-3} and 1 M, respectively). In this case, this relatively high concentration of structural defects is responsible for the significant increase of the N_2 uptake.

Further, in order to check the influence of trapped Cl^- content on the proton conduction performance, a sample with a Cl/Zr ratio of 3/7 (after Soxhlet extraction with water for three days), against 1/1 for the reference sample, was tested. The proton conductivity of this sample under the same testing condition reported in the main text decreased to $2.7 \times 10^{-3} \text{ S cm}^{-1}$, which supports the critical importance of the NH_3^+/Cl^- presence in the pore to the corresponding proton conduction behavior of the MOF.

A sentence to explain the nitrogen adsorption isotherm and pore size profiles for the treated samples was added in the revised main text (line 198-203). All the related characterization results and analyses were included in the updated Supplementary Information (Supplementary Figs. 7, 8 and 11; Tables 2-4; Note 3).

“2. In connection with the above, pore size distribution analysis should be carried out to better characterise the framework. The only information on this aspect is derived from calculations, but experimental evidence is also needed. The N_2 isotherms available should be enough to determine pore size distribution down to about 10 Angstrom diameter, which could already be useful to see differences before and after treatment.”

Response:

Pore size distribution analysis was carried out for all the investigated samples and the corresponding figures were added in the revised Supplementary Information (Supplementary Fig. 8). This analysis clearly revealed that most of the considered treatments led to a significant increase of the pore size as compared to the pristine solid. This information is added in the revised main text (line 198).

“3. Evident differences also in the TGA curves arise after treatment, but the way the curves are currently drawn does not help to appreciate them. I recommend to plot the TG curves normalising to the formula weight of ZrO_2 as 100%, which is the likely decomposition product formed at 600 degrees, and to discuss these results more carefully.”

Response:

We have re-collected the TGA data for all the samples involved from room temperature to 800 °C in oxygen with a heating rate of 2 °C/min. The corresponding TGA curves were included as Supplementary Fig. 11 and related calculations to illustrate the difference between each sample were added as Supplementary Table 3 and Supplementary Note 3 in the revised Supplementary Information.

“4. The authors determine the Cl/Zr ratio using EDX, whilst they assume that the Cl/N ratio is 1. I believe that this is not sufficient to have a correct chemical knowledge of the compound and elemental analysis should be performed to determine the CHN content.”

Response:

Elemental analysis of C, H, N content was carried out for all the investigated samples. The corresponding results were included as Supplementary Table 4 in the revised Supplementary Information.

“4. Minor points are the following:

In the introduction (Page 3, Lines 51-54) the authors state “Nevertheless, so far, the reported MOFs with high proton conduction are hardly environment-friendly to fulfill the sustainable development criteria, due to the involvement of either toxic metal ions or time and effort consuming organic linker synthesis.”. References should be added to support this statement.”

Response:

The references corresponding to this statement were added as Ref 4-11 in the revised main text.

“In the caption for Figure 1, the SBU formula is wrongly reported as “Zr₆(μ₃-O)₄(μ₃-OH)₄(COOH)₁₂”. The carboxylic groups should be deprotonated in order to ensure electroneutrality. Please correct.”

Response:

We apologize for this mistake. The reviewer is right. We have corrected accordingly the formula to Zr₆(μ₃-O)₄(μ₃-OH)₄(COO⁻)₁₂ in the revised main text.

“At Page 5, Line 105, it stated that the synthesis of most Zr-MOFs is carried out in solvothermal conditions. I do not agree with this terminology, because Zr-MOFs are usually synthesized at temperatures below the normal boiling point of the amide solvent. Since no autogeneous pressure is developed, this does not classify them as properly solvothermal.”

Response:

We have deleted all the ‘solvothermal’ in the revised main text and changed them into ‘carried out in sealed reactors’ (line 105) and ‘harsh conditions’ (line 110).

“At Page 7, line 136, the expression “under synchrotron beamline” should be changed to “under synchrotron radiation”.”

Response:

We have changed it to ‘under synchrotron radiation’ in the revised main text (line 137).

“At Page 7, Line 152, it is stated “The uncoordinated amino groups are highly distorted over four positions.” The term “distorted” should be changed into “disordered”.”

Response:

We have changed the term ‘distorted’ by ‘disordered’ in the revised main text (line 153).

Reviewer #2 (Remarks to the Author):

“The authors report a green and efficient method to synthesize the MIP-202 (Zr) constructed from natural α -amino acid. The Zr-MOF shows outstanding proton conductive performance. Besides, the authors also use quantitative analysis to explore the mechanisms. These features bestow this novel material with good application potential. Therefore, I would like to recommend the publication of this manuscript with the following revisions.”

We thank the reviewer for his/her positive evaluation of the paper.

“Major revision is suggested.

1. Line 116, the authors claim “This procedure also results in a high space-time yield of 7030 kg m⁻³ day⁻¹ comparable to some industrial scale syntheses”. However, no scalable preparation results could be found in this paper. More evidence should be provided since the large-scale production process might have a big difference with the lab-scale preparation.”

Response:

A scale-up synthesis that produced more than 30 g of sample was carried out. The calculated space-time yield of this reaction is 7296 kg m⁻³ day⁻¹. We have added the detail of this scale-up reaction in the revised main text (line 331-333).

“2. Line 203, the authors point out that the formation of orientated H-bond network in MOFs. Please provide some experimental evidence.”

Response:

We are sorry for the vague wording that causes the misunderstanding here. In this section, we did not claim that we experimentally revealed the formation of orientated H-bond network in MIP-202(Zr), but rather that the overall feature of this MOF is expected to favor a H-bonded network. This assumption was further supported by our computational effort. Therefore, the sentence in the previous main text was reconsidered to avoid any confusion as shown in the revised main text (line 212-214).

“3. Line 204, the authors claim “a good hydrolytic stability to ensure that the proton conductive performances can be maintained over cycles.” Could the material keep stability under redox conditions? Maybe Fenton’s test should be conducted.”

Response:

The stability of the sample under the Fenton reaction condition was tested. As it is well-known that destructive ability of the Fenton reagent depends on the concentrations of H₂O₂ and Fe(II) salt, various H₂O₂ concentrations were tested. The MIP-202(Zr) sample became amorphous in less one hour when it was soaked in 50%wt H₂O₂ without adding any Fe(II) salts. It suggested that MIP-202(Zr) is not stable under strong oxidative condition. However, MIP-202(Zr) displays much better resistance when 5%wt H₂O₂ was used with equivalent (NH₄)₂Fe(SO₄)₂·6H₂O. As shown in the Figure R1, the PXRD pattern of the MIP-202(Zr) sample treated for 24 hours showed clear peak broadening thus decreased crystallinity could be concluded in comparison

with that of the pristine sample. Therefore, MIP-202(Zr) degrades slowly under mild oxidative conditions, however the sample is not stable under strong oxidative conditions.

Figure R1. PXRD comparison of MIP-202(Zr) samples before and after the treatments under different oxidative conditions.

“4. Quantitative analysis are used to explore the mechanism of high proton conductivity of MIP-202 (Zr). The simulation results would become more convincing if they are verified or partially verified by experimental results. “

Response:

We do agree with the reviewer. The Grotthus-like proton conduction mechanism evidenced by molecular simulations was supported by the value of the activation energy E_a estimated from our conductivity experiments performed at different temperatures. Indeed, the resulting E_a of 0.22 eV is significantly lower than 0.40 eV, which is commonly associated with a Grotthus-like mechanism for proton transport.

“5. Line 282, the authors claim “The adsorbed water molecules are only able to form small discontinuous aggregates of hydrogen-bonded clusters within the pores of MIP-202(Zr)”. Will these discontinuous aggregates hydrogen-bonded clusters confine the long-range proton conduction?”

Response:

To avoid the ambiguity that the reviewer commented on, the corresponding section was revised as shown in the updated main text (lines 297-302).

Reviewer #3 (Remarks to the Author):

“The work reports the synthesis, structural stability study, proton conductivity, and MC results of a MOF. The MOF was stable under several solvent conditions and acidic or basic media. This material was also prepared under mild reaction conditions. Proton conduction of the MOF was investigated experimentally and theoretically. However, the conductivity was just as high as 0.01 S cm^{-1} , which is not significant compared to the state-of-the-art conductivity ($>0.1 \text{ S cm}^{-1}$) observed in several MOFs. Therefore, I think this paper is not suitable for this journal. The following concerns should be addressed.”

Response:

We agree with the reviewer that the proton conductivity of MIP-202(Zr) is not record-breaking. However, we would like to emphasize that MIP-202(Zr) is, to the best of our knowledge, the first proton conductive MOF that shows high performance while following the green chemistry principles and thus sustainable development requests.

A very limited number of MOFs (less than 10) displaying proton conductivity at this level ($\geq 0.1 \text{ S cm}^{-1}$) have been reported so far. Among them, only BUT-8(Cr)-SO₃H showed comparable performances under similar conditions. As we commented in the main text and summarized in the supplementary information, all these MOFs with high proton conductivity were prepared under harsh conditions, and/or using harmful and complex chemicals, which introduces significant inconvenience for further scale-up synthesis at a practical level.

Therefore, we are strongly convinced that MIP-202(Zr) is amongst the most attractive MOFs as water-mediated proton conductor.

“1. The dinitrogen sorption data (Fig. S3) showed that all treated samples have adsorption capacities greater than as-made sample. This indicates that the chemical treatments affect the porosity in spite of structural integrity. The authors should discuss what happens.”

Response:

As it is basically the same comment as the one from the first reviewer, please see our response to the first comment of reviewer 1.

“2. To support the Grotthus mechanism, additional experiments were added to manuscript.”

Response:

We identified the Grotthus mechanism through (1) the evaluation of the activation energy, obtained from ac impedance data, which gives a value that ranges in the interval 0.10 eV - 0.40 eV commonly attributed to such a mechanism (ref: Padmini Ramaswamy, Norman E. Wong, Georges K.H. Shimizu, Chem. Soc. Rev. 2014, 43, 5913-5932) and (2) the predicted water arrangements that are optimal to favor such a mechanism.

“3. Calculate the yield of MIP-202(Zr)”

Response:

The reaction yields were added in the revised main text (line 329 and 332).

“4. Check typos in the manuscript. For instance, Table 1 in line 112 should be changed to Table 2.”

Response:

We have checked typos and corrected them in the revised manuscript.

REVIEWERS' COMMENTS:

Reviewer #1 (Remarks to the Author):

The authors have carefully considered all the points raised by the reviewers. It is still surprising to me that after alkaline treatment the Cl/Zr ratio is about 0.72, but the pore size is clearly increased from 3.7 to 5.2 angstrom, which would suggest massive removal of Cl from inside the framework. This could be due to Cl migrating from being a simple counteranion for ammonium to coordinating to the metal clusters, but figuring this out would require a considerable effort that goes beyond the scope of this work.

Overall, I believe that the manuscript is now suitable for publication